

# Investigation of Effects of Coronal Mass Ejections on Ionospheric Total Electron Content over Nsukka, South Eastern Nigeria

Esther A. Hanson[1,2], F. Nneka Okeke[1], Kingsley Okpala[1]

[1]Department of Physics and Astronomy, University of Nigeria, Nsukka, 410001, Nigeria
[2]Advanced Space Technology Applications Laboratory, University of Uyo, Uyo, 520271, Nigeria

*Correspondence to*: Esther A. Hanson (esther.hanson.pg78720@unn.edu.ng)

**Abstract.** In this work, we attempted to investigate the contributions and effects of coronal mass ejections (CMEs) on total electron content (TEC) in the ionosphere of an equatorial station, Nsukka (Lat. 6.86N; Long 7.38E) located in South Eastern Nigeria. Using TEC data recorded by the Global Positioning System (GPS) of the Air Force Research Laboratory, USA, and CME data obtained from the USA owned Solar and Heliospheric Observatory (SOHO) satellite, we calculated the variation of TEC in the solar maximum year 2012, and observed regular, Gaussian distribution of TEC during geomagnetic solar quiet

(Sq) days. On days associated with CME events, TEC variations assumed very sporadic patterns; maximized quite early during geomagnetic disturbed days and peaked at later hours during quiet days. The ionospheric electron contents are generally very low at both pre-noon and nighttime hours but quite high at noon and post-noon hours. This pattern of TEC variation is due to fluctuation in solar radiations incident on earth's equatorial ionosphere. During quiet periods the number of free electrons generated is smaller in comparison to that generated during disturbed times, which shows a positive contribution of CMEs to

TEC profile. TEC profiles for days categorized as neither disturbed nor quiet are synonymous to TEC curves on quiet days. There is significant time-variance in peak-time of TEC between disturbed and quiet days. TEC maximized quit earlier on disturbed days, implying increased influx of charged particles into the ionosphere due to the prevailing CME events. These results can contribute as input to the ionospheric monitoring and forecasting for the equatorial region of South Eastern Nigeria.


## 1 Introduction

Modern society depends heavily on a variety of technologies that are susceptible to the extremes of space weather — severe disturbances of the upper atmosphere and of the near- Earth space environment that are driven by the magnetic activity of the Sun. Strong electrical currents driven in the Earth's surface during auroral events can disrupt and damage modern electric

power grids and may contribute to the corrosion of oil and gas pipelines. Changes in the ionosphere during geomagnetic storms driven by magnetic activity of the Sun interfere with high-frequency radio communications and GPS navigation (Jakowski et



al., 2011). Space explorers must be constantly aware of the current space weather and be prepared to handle the most extreme conditions that might be encountered (Andrews, 2003).

The ionosphere is that region of the atmosphere from about 60 to 1000km above the earth's surface, where free electrons exist in numbers sufficient to influence the transmission of electromagnetic waves at radio frequencies. Hence, the ionosphere could

be regarded as an ionized region of the upper atmosphere, which forms the inner boundary of the magnetosphere. The main reason for the existence of the ionosphere is the absorption of solar radiation by the upper atmosphere, thereby dissociating the molecules and liberating free electrons. In any part of the ionosphere, the electron density $N_e$ is determined by the equation of continuity for electrons;

$$\frac{\partial N_e}{\partial t} = P(N_e) - L(N_e) - \nabla.(N_e V) \qquad 1$$

where $\frac{\partial Ne}{\partial t}$ = time rate of change of electron density, P(Ne) = rate of production of electrons, L(Ne) = loss of electrons (mostly through chemical processes), Div(Ne) = rate of outward transportation of electrons (i.e. sink), V = mean velocity of electrons. Mukherjee et al., (2010) studied the variability of TEC over the crest of equatorial anomaly station in Bhopal during solar activity period (2005 – 2006) using GPS and observed a higher TEC variability on quiet days as compared to disturbed days. Tsai et al., (2001), Oko et al., (2012) and Okala et al., (2013) reported a seasonal variation of ionospheric total electron content

in equatorial anomaly regions, pointing out that these phenomena could be fully explained by a combined theory of the transequatorial neutral wind, the subsolar point and the equatorward wind. These variations are characterised by maximum around the noon or post-noon, and minimum at night and early morning periods (Oko et al., 2012). At low latitude ionosphere the diurnal behaviour of scintillation is driven by the formation of large-scale equatorial depletions (Olwendo et al. (2012), which are formed by post-sunset plasma instabilities via the Rayleigh–Taylor instability near the magnetic equator.

Though halo CMEs, pointed along the Sun-Earth-line, are known to be the main drivers of space weather disturbances, Schwenn et al. 2005, Gopalswamy, 2005 and Hundhausen et al. 2012 noted that 15% of comparable cases does not cause any ICME signature on Earth at all. This work aims at using GPS data generated by the United States Air Force Research Laboratory to investigate possible effects of Coronal Mass Ejections (CMEs) on ionospheric electron content (TEC) over Nsukka (Lat. 6.86N; Long 7.38E) for a solar maximum year, 2012.


## 2 Theories of CMEs Propagation and Estimation of TEC

It was first postulated that CMEs might be driven by the heat of an explosive flare. However, it soon became apparent that many CMEs were not associated with flares, and that even those that were associated it often started before the flare. Because CMEs are initiated in the solar corona, which is dominated by magnetic energy, then their energy source must be magnetic.

Because the energy of CMEs is so high, it is unlikely that their energy could be directly driven by emerging magnetic fields in the photosphere (although this is still a possibility). Therefore, most models of CMEs assume that the energy is stored up in the coronal magnetic field over a long period of time and then suddenly released by some instability or a loss of equilibrium



in the field. There is still no consensus on which of these release mechanisms is correct, and observations are not currently able to constrain these models very well.

The estimation of TEC using GPS receivers is made possible by the dispersive nature of the ionosphere (Tsai et al., 2001; Mukherjee et al., 2010). Signal delay caused by TEC in the ionosphere is corrected in the dual-frequency measurement by a
linear combination of L1 and L2 band frequencies (Jakowski et al., 2011). The ionosphere imparts a group delay (D) to a radio frequency (RF) signal that are equal in magnitude and proportional to the total number of electrons encountered along the line of sight (LOS) and inversely proportional to the square of the signal frequency as expressed in equation (2)

$$D = \frac{kTEC}{f^2} \qquad\qquad 2$$

where, D = Group delay in seconds (s), K = constant of proportionality = $40.30 m^3 s^{-1}$, f = signal frequency per second, TEC =
total electron content (electrons/$m^3$). TEC corresponds to the number of electrons contained in a column of unitary base that extends from earth surface to a determined height in the atmosphere; say, the ionosphere. It is expressed as; 1 TEC Unit =1 x $10^{16}$ e/$m^2$.

## 2.1 Method of Analysis for TEC Enhancement

This study employs an empirical method of identifying possible enhancements of TEC by CME. We stipulate that an increase in the amplitude of TEC level above its solar quiet-time-level ($TEC_{sq}$) is an indication of CME's contribution to enhancement in ionospheric TEC, where all other possible contributions are assumed to be negligible. Thus,

$$TEC_{CME} - TEC_{sq} = \Delta TEC \qquad\qquad 3$$
$$\Delta TEC > 0 = Enhancement \qquad\qquad 4$$

where $TEC_{CME}$ = hourly TEC value on CME-event-day, $TEC_{sq}$ = hourly TEC value on solar quiet day,     $\Delta TEC$     = difference between $TEC_{CME}$ and $TEC_{sq}$.

## 2.2 Method of Analysis for Correlation between TEC Enhancement and Speed of CMEs

In the analysis of data for this project, statistical correlation has been applied. Correlation is a measure of association between two variables. The Spearman's rank correlation coefficient ρ is given by;


$$\rho = 1 - \frac{6 \sum d_i{}^2}{n(n^2 - 1)} \qquad\qquad 5$$

Where $n$ is the number of (XY) pairs, $d$ is the difference between each pair of ranks, $\sum d^2$ is the sum of the squared values of $d$.

## 3.0 Sources of Data

The data deployed in this study was obtained from various sources. CME data was retrieved from the CME Catalog of NASA's
Large Angle Spectrometric Coronagraph Experiment on board the Solar Heliospheric Observatory (SOHO/LASCO), while TEC data was provided by the United States Air Force Research Laboratory.



## 4.0 Analysis of Data and Discussion of Results

### 4.1 Data Analysis

In this section we analysed TEC profiles on geomagnetic quiet days, when solar activity is approximately negligible. Figs. 1, 2 and 3 depict TEC variations for the first two quietest days in the month of February, namely February 17, 23 and the

geomagnetic quietest day in the month of March, namely March 26, 2012.

TEC data for March 19, April 10, May 21 and May 28, in year 2012 was analysed to determine TEC profile in the study area for days categorized as CME-free, non-solar-quiet and non-disturbed days. This constitute days with three composite attributes; that neither had CME events, nor described as disturbed, nor described as solar quiet. Fig. 4 shows TEC profile of March 19, 2012. Figs. 5 – 7 depict the variations of TEC on April 10, May 21 and May 28 respectively.

TEC computation was carried out for March 2, March 5, March 9, and March 21, May 12, and June 11, 2012, to evaluate the degree of contribution of coronal mass ejections to the total electron content in the Nsukka ionosphere during the period under investigation. Figs. 8 – 13 show plots of hourly variation of TEC on both geomagnetic quiet days and days of CME events, in attempts to compare the responsiveness of TEC to CMEs. TEC profile of March 26, 2012 is used as base-line for the evaluation of the degree of TEC enhancement on March 2, March 5, March 9, March 21, May 12 and June 11, 2012. On the other hand,

TEC profile of February 17, 2012 serves as a base-line value for calculating TEC enhancement for May 12, 2012. And these base-lines have been used in calculating TEC enhancements with respect to the peak amplitude of TEC on CME –event-day. Spearman's correlation analysis was carried out using equation (5), to determine the relationship between CME speed and TEC enhancements.

### 4.1 Discussion of Results

Fig. 1 shows the variation of TEC on the quietest day of the month of February, February 17, 2012. On this quietest day, TEC increased steeply from 18TECU at 07:00 hours to about 42 TECU at noon and maintained a quasi-plateau till about 15:30 hours, where an onset of decline was observed. The magnitude of TEC reduced from 35TECU at 18:00 hours to about 2TECU at 22:00 hours. Here, the TEC variation on the quietest day indicates the normal enhancement of free electrons in the ionosphere

as solar radiations ionize atoms and molecules. The ionization increases as the Sun rises above the horizon and decreases as the Sun sets.

Fig. 2 depicts ionospheric TEC variation on the second quietest day of the month of February, namely February 23, 2012. At 08:00 hours TEC values recorded 19 TECU and followed a similar trend observed in Fig. 2 where 42 TECU electrons were recorded at 12:00 hours. Contrastingly, TEC reached a peak value of $45 \times 10^{16}$ em$^{-2}$ at 14:00 hours. Also, the time- dependent

ionospheric TEC distribution assumes a Gaussian trend with a peak at 14:00 hours, signifying highest solar ionization at a post-noon hour. This agreed with Adekoya et al., (2015).

Fig. 3 shows TEC variation for the geomagnetic quietest day in March (i.e. March 26, 2012). From the recorded data, at 07:30 hours electron content reached 18TECU and increased gradually to 45 at noon, where it remained steady till 13:00 hours.




Thereafter, it increased steeply to 54 TECU at 15:00 hours and declined steeply to 42 TECU at 18:00 hours and subsequently to 17 TECU at 20:00 hours. TEC varied sporadically during the late night-time-hours, averaging about 17TECU.

Fig. 4 depicts the ionospheric TEC profile of March 19, 2012 – categorized a non-CME, non-solar-quiet and non-disturbed day. From the profile, electron content was 18 TECU at about 0800 hours (LT). The maximum ionization of electrons reached
54 TECU at noon. It declined to 50 TECU at 1300 hours and remained constant till 18:00 hours, where it dropped sharply to 20 TECU at 2000 hours. The curve in Fig. 5 shows the variation in electron content across the day for April 10, 2012, with a peak of 47 TECU at 1600 hours. The Gaussian curve thus projects a pattern of ionization devoid of contributions from CMEs. Fig. 6 is the variation of TEC on May 21, 2012, possibly without the influence of CMEs. TEC increased from 26TECU at 0800 hours to a peak of 53 TECU at 1300 hours. It gradually decreased to the range of 45 and 35 TECU at post-noon hours.
TEC finally decreased to an average of 20 TECU during nighttime hours. The profile curve in Fig. 7 depicts the ionospheric electron content on May 28, 2012, from 0730 to 1600 hours. TEC increased uniformly from 17 TECU at 0730 hours to 43 TECU at 1330 hours. A slight increase was observed between 0330 and 1430 hours. Subsequently, it decreased to 42 TECU at 1600 hours. The gradual decrease observed in TEC during post-noon and nighttime hours (as observed in Figs. 4 – 6) is due to recombination of electrons resulting from reduced solar radiation as the Sun sets beyond the horizon.

In Fig. 8 TEC amplitude increased gradually from 30 TECU and peaked at 55.11 TECU on March 2, 2012, while it peaked at 52 on the quiet day of March 26, 2012. TEC peaked at 60.54 and 67.67 on 5[th] and 9[th] March respectively (see Figs. 9 and 10). There were overlaps in TEC magnitudes on both disturbed and quiet days as observed in Figs. 8 – 13. In Fig 8, an overlap occurred at 1600 and 1800 hours. Nonetheless, in Fig. 9 an overlap is evident at 1600 hours. In Figs. 10, 11, and 12, overlaps occurred at 0800 hours, 2200 hours 15 hours respectively. Whereas, in Fig. 13 TEC magnitudes for disturbed and quiet days
intersected at two points namely 44 and 20 TECU, which correspond to 1200 and 1900 hours respectively.

A trace of the curve in Fig 11 would reveal that TEC amplitude on the quiet day (i.e. March 26, 2012) out-weighed that of disturbed of March 21, 2012. In Fig. 13, TEC trend of CME-event day (i.e. June 11, 2012) maintained a constant value of 45.03 from about 11:00 hours till 18:00 hours, where it declined to 20 TECU at 19:30 hours and averaged 12 TECU during nighttime hours. This was much exceeded by the quiet-time values, which peaked at 53 TECU at 15:30 hours. This agrees
with Mukherjee et al. (2010) who studied the variability of TEC over the crest of equatorial anomaly station in Bhopal during solar activity period (2005 – 2006) using GPS and observed a higher TEC variability on quiet days as compared to disturbed days.

Also observed was significant time variance between peak-time of TEC on CME-event-days and on quite days. TEC maximized quit earlier on days associated with CMEs. In Figs. 8, 9, 10 and 12, crests of the undotted curves, (which are TEC
amplitudes on CME-days) maximized at approximately 1400 hours. On the other hand, the dotted curves maximized at 1500 hours.

TEC enhancements were obtained by subtracting TEC values on a geomagnetic quiet day, $TEC_{sq}$, from TEC values on CME-event-day, $TEC_{CME}$. On March 2, TEC amplitude attained a maximum value of 55.11 TECU at 13:30 hours, whereas, at this time on the quiet day, TEC was observed to be 46.95 TECU, resulting to an enhancement of 8.16 TECU. Similar procedure



was adopted for calculating TEC enhancements for March 5, March 9, March 21, May 12 and June 11, 2012 (see Figs 8 – 13). And from evaluations, 9.08, 18.48, -5.38, 5.87 and -8.41 TECU were obtained as TEC enhancements for the afore-mentioned days respectively. It is interesting to note negative enhancements for March 21 and June 11, 2012, signifying higher values of TEC on the quiet day in comparison to magnitude of TEC on CME-event-day. This calls for a deeper insight to research on

defining parameters for solar quietness.

The Spearman correlation coefficient shows that there is a weak correlation between CME speed and TEC enhancement. From calculation, the correlation coefficient between CME speed and TEC enhancement is 0.2. This implies that the speed of CME may not necessarily be the sole determinant factor to TEC enhancement. Other CME parameters such as the number density could be propelling the increment of electrons in the ionosphere during CME events.

## Conclusions

The ionospheric electron contents are generally very low at both pre-noon and nighttime hours but quite high at both noon and post-noon hours. This pattern of TEC variation is due to fluctuation in solar radiations incident on earth's equatorial ionosphere. During quiet periods the number of free electrons generated is smaller in comparison to that generated during disturbed times.

TEC profiles for days categorized as neither disturbed nor quiet are synonymous to TEC curves on quiet days

There is significant time-variance in peak-time of TEC between disturbed and quiet days. TEC maximized quit earlier on disturbed days. These results can contribute as input to the ionospheric monitoring and forecasting for the equatorial region of South Eastern Nigeria.

**Acknowledgements**

We are grateful to Dr. Keith Groves of the United States Air Force Research Laboratory for technical support and making the PGS TEC data accessible for this research through the collaboration between the Centre for Basic Space Science (CBSS), University of Nigeria and the USA Air Force Research Laboratory. We are also appreciative of the goodwill of the Director of CBSS, University of Nigeria, Nsukka, Prof Pius N. Okeke for forging the collaboration between Nigeria and USA on the

SCINDA project.

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

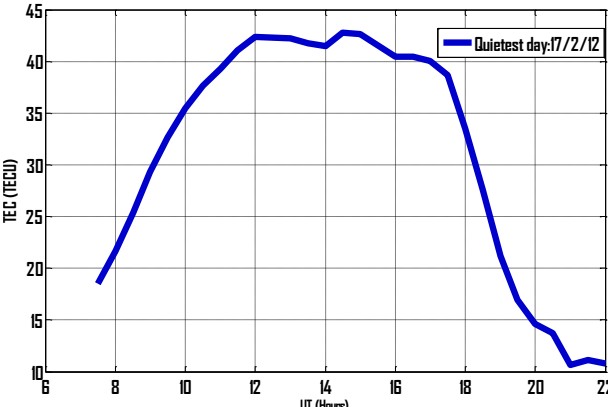

**Fig. 1:** TEC profile of Feb. 17, 2012 for Nsukka (Q1)

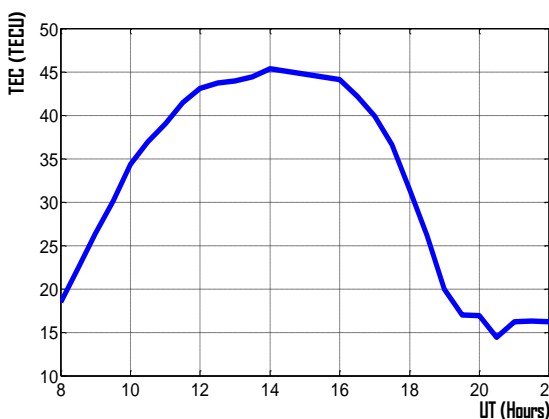

**Fig. 2:** TEC profile of Feb. 23, 2012 for Nsukka (Q2).




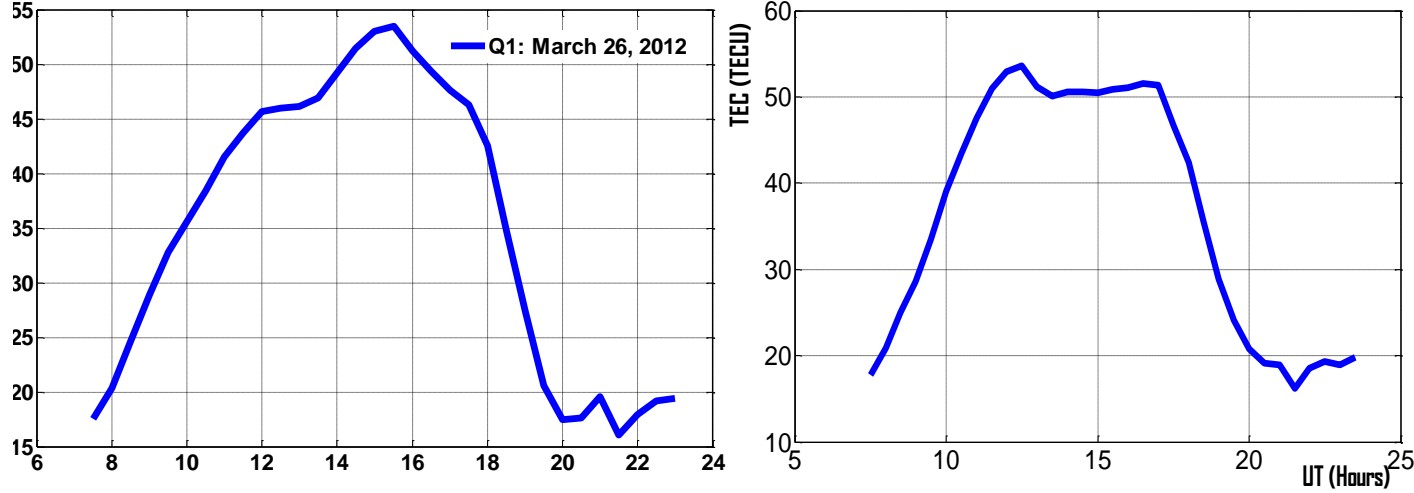

**Fig. 3:** TEC profile March 26, 2012 for Nsukka (Q1)

**Fig. 4:** TEC profile of March 19, 2012 for Nsukka

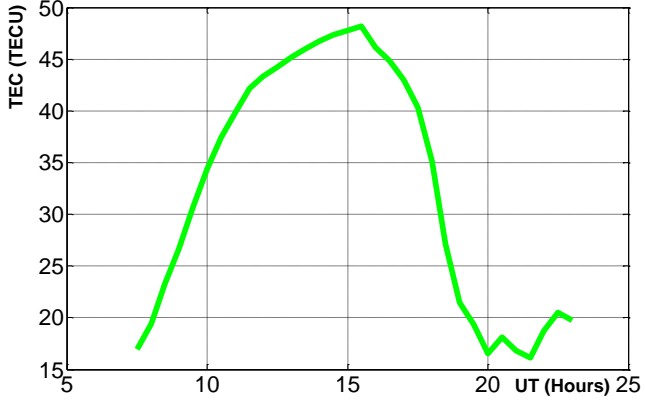

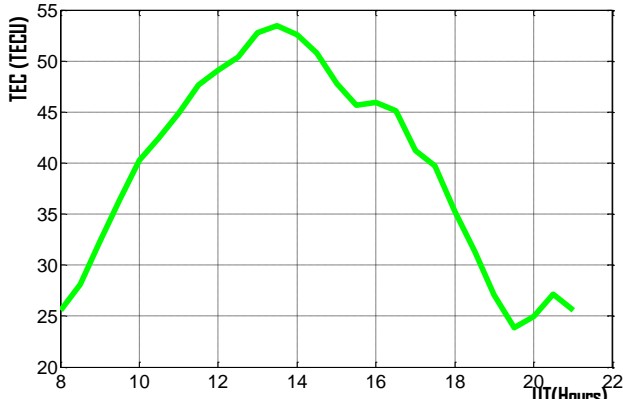

**Fig. 5:** TEC profile of April 10, 2012 for Nsukka

**Fig. 6:** TEC profile of May 21, 2012 for Nsukka



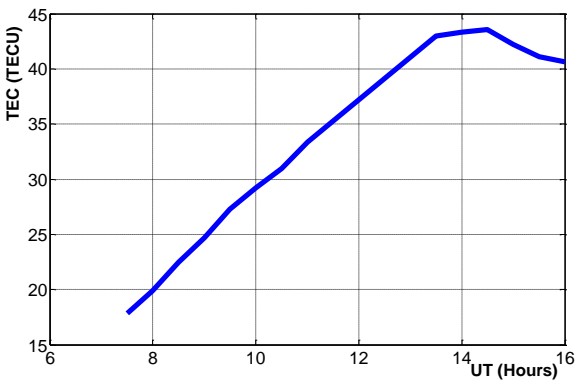

**Fig. 7:** TEC profile of May 28, 2012 for Nsukka

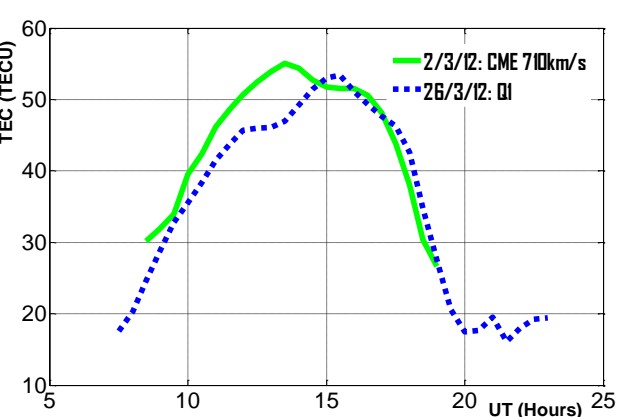

**Fig. 8** TEC variations on quietest day and Mar. 2, 2012.

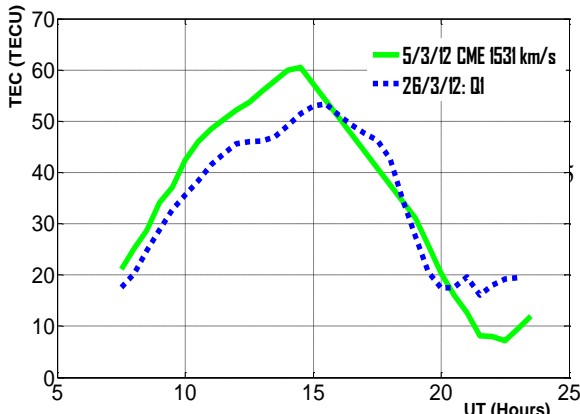

**Fig. 9:** TEC variations on quietest day and Mar. 5, 2012.

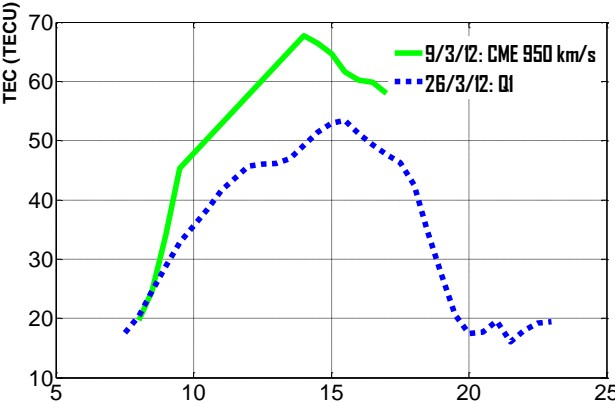

**Fig. 10:** TEC variations on quietest day and March 9, 2012. 201

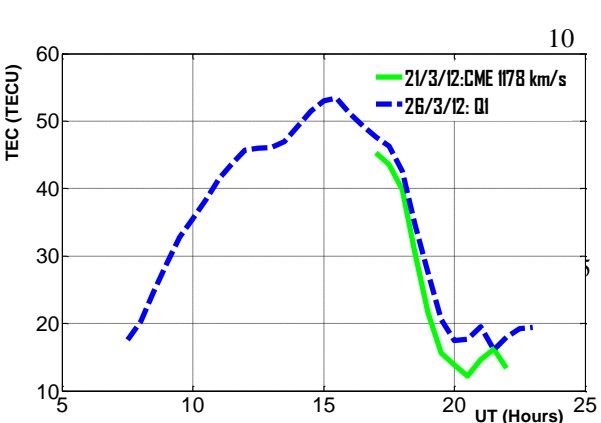

**Fig. 11:** TEC variations on quietest day and March 21, 2012.

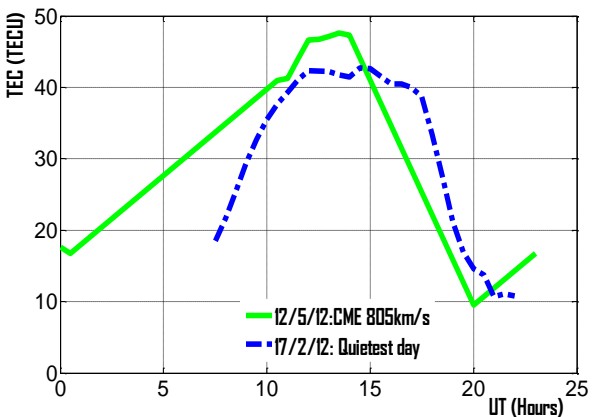

**Fig. 12:** TEC variations on quietest day and May 12, 2012





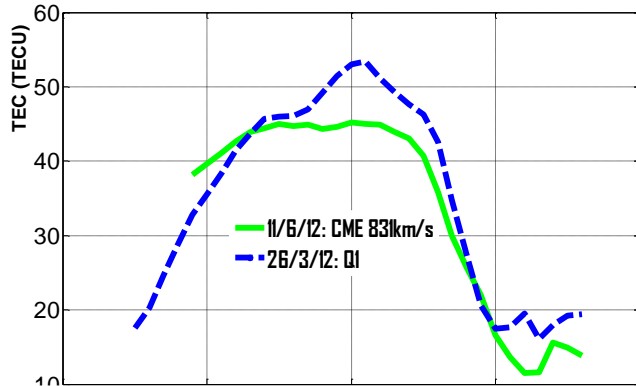

**Fig. 13:** TEC variations on quietest day and June 11, 2012.