# Peer review of "Investigation of Effects of Coronal Mass Ejections on Ionospheric Total Electron Content over Nsukka, South Eastern Nigeria"

_Annales Geophysicae, 2019_

## Referee Comment (RC1) · Anonymous Referee #1 · 22 May 2019

Comment on "Investigation of Effects of Coronal Mass Ejections on Ionospheric Total Electron Content over Nsukka, South Eastern Nigeria"

This manuscript reports the GPS/TEC observations over Nsukka, in south eastern Nigeria. The authors focus on the diurnal variation of the GPS/TEC and discuss its connection to the speed of the CME from the Sun. The study, however, does not mention the physical processes behind the ionosphere-CME connection at all. If the causal effect is explained in the framework of the magnetic/ionospheric storm, the CME must hit on the magnetopause first, then cause geomagnetic disturbances, and by changing the electric field or injecting heat into the atmosphere to disturb the ionosphere.

My concerns are therefore as follows: 1. The authors did not show how they choose the CME events. Did these CME actually hit on the magnetosphere and cause geomagnetic disturbances? If not, how can they affect the ionosphere? 2. The diurnal variation of the ionosphere can be affected by several mechanisms related to the coupling between the magnetosphere, ionosphere, and the thermosphere. For example, the prereversal enhancement (PRE) of the electric field in the ionosphere, is known to occur around sunset and shortly enhance the ionospheric electron density before night. This feature seems to be exhibited in some of the authors' figures, but it is interpreted as an effect of CME in the text. 3. The data used in Fig. 10 and Fig. 12 seem to have some gaps (noticing the straight green line segments in the figures). The gaps could largely distort the diurnal variation of the TEC, so the data should be discarded or replaced with more continuous records.

---

## Referee Comment (RC2) · Anonymous Referee #2 · 22 May 2019

Referee #2

My comments on the paper ANGEO-2019-39, Investigation of effects of coronal mass ejections on ionospheric total electron content over Nsukka,south eastern Nigeria by A.A. Hanson, F.N. Okeke, K. Okpala

1. No list of CME and the main identification parameters (for example: Sun emission date, probable arrival time in the vicinity of the magnetosphere, terrestrial impact via the Dst index, etc.) is not present: **make a comprehensive table**
I simply note that a speed is present on the figures without any indication in the text on its origin: **insufficien**

**t**
2. Too little clarification on the calculation of VTEC (V = vertical) by the SCINDA receiver.
- How is VTEC obtained from STEC (S = Slant)? [I know the method but the authors do not address the subject and the problem of calibration of bias is important for absolute values].
- What is the rate of the measurements (1 pt per mn?)
- No inventory of 2012 measures It is important because we will see that it has a lot of interruptions in the series of measures and that not all CMEs are covered: **make a summary table**

3. With the observation of figures 1-13, it seems to me that there are many time periods without GPS measurements:
- The curves start at the beginning of the day, at different times (no nighttime values);
- Some curves are linear over several hours (Figure 10 between 9h and 14h for example): the 2 endpoints were connected by a straight line (?) which is not physical. It lacks a presence symbol of the measure.

4. No selection criterion between quiet day and disturbed day: Knowing the time of the CME ejection and an estimate of the mean speed of propagation in interplanetary space (see the data of ACE satellite in OMNIWEB, https://omniweb.gsfc.nasa.gov/form/sc_merge_min1.html).
On which magnetic index, the authors decide that it is 1 (or more) disturbed day?

5. In paragraph 2.1, the authors describe their work quickly
- On the search for enhancement of VTEC. Why only increases and not decreases (which they will observe later in some cases);
- The treatment is announced on hourly values: Is the round hour, an average over an hour? knowing that the comments in the text will involve half-hours(page 4, line 22 for example);
- The figures should connect the 24 time points. However, slope breaks indicate points (not present by a symbol) closer together: Put the approaches in coherence;
- are the enhancements to be taken into account all day long? No clarification from the authors on a minimum duration to choose an increase of the TEC: **to clarify**

6. On the set of figures 1-13.
- The 13 figures are drawn with a single identical model, which is not acceptable for many figures. Fortunately, the authors treat only a few cases, otherwise we would have hundreds of figures! Be imaginative and original on the graphs!

- The problem of connecting time points (no symbol reported) is not physical: I want to see your measurements on the graphs!
- The annotation 'TEC (TECU)' is not homogeneous in position, size and font;
- Adopt a single scale of VTEC variation that will highlight the seasonal variation that is currently visually masked;
- The scale of the abscissas is in UT hours: I would have liked a scale in local time to better identify the different phases in the day even if the difference is small (~ 30 min for 7.38°E longitude);
- 'TEC profile'?, The word 'profile' is rather reserved for the variation in altitude of the ionization.

7. The writing of the bibliography is correct (correct Valladares instead of Valludares).

8. In summary
The authors have an irregular series of TEC measurements at an original position for an equatorial ionosphere (at this point the magnetic position is absent from the text). Exploitation work may be published. However, from my point of view, the work presented is not enough to be published. Faced with a real lack of measurements (electrical power cuts?) and the receiver is a SCINDA scintillator, I advise authors to add a study scintillation indices (S4 and sigma-phi) that will complement their TEC study and consolidate the contents. The authors will then be able to integrate their results with the NIGNET network measures for other publications.
**I reject the manuscript to the publication (Not acceptable)**

---

## Author Comment (AC1) · 6 Jun 2019

RESPONSES TO QUERIES FROM ANONYMOUS REFEREE #1

Yes, when CME is directed towards the Earth as an interplanetary CME, the shock wave of the traveling mass hits the magnetopause and causes a geomagnetic storm which may result in disruption of the magnetosphere; where the magnetosphere is compressed on the day-side and extended on the night-side magnetic tail. Since all these are well known theories, we decided not to bother the reviewer with all these details. However, we can include it. 1.The selection criteria for CMEs were; (i) high speed CMEs, (ii) earth-directed CMEs which

were proven to have caused geomagnetic storms. For instance, Fig. 10 showed an increase in TEC on March 9, 2012. As a validation of our data selection, NASA reported an auroral event and geomagnetic storm occurrence on March 8, 2012 (see https://www.nasa.gov/mission_pages/sunearth/news/News030712-X5-4.html. Earlier, EarthSky reported an X5-class solar flare events occurred on March 6-7, 2017 (See https://earthsky.org/space/another-major-solar-flare-during-night-of-march-6-7-2012 2. The diurnal variation in this study showed the contribution of energy transfer resulting from CMEs. Hence, marked contrasts are observed between TEC signatures at quietest days and those on the selected disturbed days which were day associated with CMEs and geomagnetic storms. 3. I had missing data for Fig. 10 and Fig. 11. I will discard Fig. 11. Some information could still be drawn from Fig. 10 since the data scaled beyond the peak values of TEC at both disturbed and quiet conditions. Nonetheless, I can discard it.

Esther A. Hanson Corresponding Author

Please also note the supplement to this comment:
https://www.ann-geophys-discuss.net/angeo-2019-39/angeo-2019-39-AC1-supplement.pdf

---

## Editor Comment (EC1) · Ana G. Elias (Editor) · 22 Jul 2019

Dear Dr. Hanson, On the basis of the comments received by two reviewers, your manuscript "Investigation of Effects of Coronal Mass Ejections on Ionospheric Total Electron Content over Nsukka, South Eastern Nigeria" needs much work in order to become acceptable for publication, so I am sorry to tell you that it cannot be published in the present form. Both reviewers suggested that the manuscript should be rejected. However they both make very constructive comments and one of them considers that your work has new data and ideas. So, my suggestion to you and co-authors is to withdraw the paper first, and then resubmit it including all the suggested changes. I

am sorry for this negative decision, but I hope that you consider to improve this paper based on the reveiwers' comments and resubmit a suitable version. In this case, I will consider the revision as a new submission. I look forward your new submission. Sincerely, Ana Elias